# Artesunate regulates malignant progression of breast cancer cells via lncRNA TUG1/miR-145-5p/HOXA5 axis

Chao Yang[1]*, Yunjiang Liu[2,3]*, Lingyun Gai[1], Ziteng Zhang[1], Yanshou Zhang[1], Geng Zhang[1], Kaiye Du[4], Chao Gao[4]

1 Department of Breast Center, The Fourth Hospital of Hebei Medical University, Shijiazhuang, China,
2 Department of Breast Center, Fourth Hospital of Hebei Medical University, Shijiazhuang, Hebei, China,
3 Hebei Provincial Key Laboratory of Tumor Microenvironment and Drug Resistance, Hebei Medical University, Shijiazhuang, Hebei, China, 4 Radiotherapy Department, The Fourth Hospital of Hebei Medical University, Shijiazhuang, China

* ydsyyc@hebmu.edu.cn (CY); lyj818326@hebmu.edu.cn (YL)

## Abstract

### Background

Breast cancer continues to be a predominant cause of female mortality globally, characterized by limited therapeutic options and substantial adverse effects. Artesunate (ART), a traditional Chinese medicine approved by the FDA for malaria treatment, has demonstrated potential anticancer properties against breast cancer. However, the underlying molecular mechanisms remain incompletely elucidated. This study posits that the antitumor efficacy of artesunate may be mediated through the regulation of the lncRNA TUG1/miR-145-5p/HOXA5 axis.

### Methods

A comprehensive array of in vitro assays was employed to investigate the proposed molecular pathway, including CCK-8 proliferation assay, EdU incorporation assay, Transwell invasion assay, scratch wound healing assay, TUNEL apoptosis assay, and dual-luciferase reporter assay. Additionally, Western blot analysis, quantitative real-time PCR (qPCR), and plasmid transfection techniques were utilized to validate the findings.

### Results

The results revealed that artesunate exerted a dose-dependent inhibitory effect on breast cancer cell proliferation. This was accompanied by the down-regulation of HOXA5, WNT, β-catenin, Fizz1, and Arg-1, implicating the involvement of the WNT/β-catenin signaling pathway. Furthermore, artesunate significantly modulated the expression levels of lncRNA TUG1, miR-145-5p, and HOXA5, suggesting a mechanistic role of the lncRNA TUG1 pathway in its anticancer activity.

**Data availability statement:** All relevant data are within the paper and its Supporting Information files.

**Funding:** This work was supported by the Training Project of Clinical Medical Professionals (2023), ZF2023065, provided by the government of Hebei Province. The funders had no role in study design, data collection and analysis, decision to publish, or preparation of the manuscript.

**Competing interests:** Competing Interests The authors declare that there are no competing interests regarding the publication of this paper. The research was conducted independently, and the results presented in this study are the sole responsibility of the authors.

## Conclusions

These findings indicate that artesunate may inhibit breast cancer progression through the lncRNA TUG1/miR-145-5p/HOXA5 axis, highlighting its potential as a promising therapeutic candidate for future clinical trials in cancer therapy.

## Introduction

Breast cancer persists as a formidable global health challenge, constituting one of the most prevalent malignancies among female populations worldwide. Contemporary diagnostic modalities encompass clinical examinations including mammography, breast MRI, and ultrasonography, with histopathological biopsies serving as the gold standard for malignancy confirmation and molecular subtyping, particularly through the assessment of hormone receptor status and Human Epidermal Growth Factor Receptor 2 (HER2) expression [1]. Notwithstanding substantial advancements in therapeutic interventions, persistent challenges continue to compromise patient outcomes and quality of life [2]. Conventional treatment regimens, including chemotherapy, radiotherapy, and targeted therapies, are frequently associated with substantial adverse effects [3]. Furthermore, advanced-stage (stage IV) metastatic breast cancer remains refractory to curative treatment, with limited therapeutic options available for effective disease management [4]. Given the escalating global incidence, suboptimal treatment outcomes, and restricted therapeutic armamentarium, there exists an imperative need for the development of novel therapeutic strategies for breast cancer management.

Traditional Chinese herbal medicine has emerged as a promising complementary therapeutic approach in oncology, demonstrating potential benefits including mitigation of conventional therapy-induced adverse effects, enhancement of tumor response rates, improvement in quality of life through symptom palliation, modulation of the tumor micro-environment, and potential extension of overall survival [5,6]. Artesunate, a sesquiterpene lactone derivative isolated from *Artemisia annua*, represents a novel therapeutic agent. Following its FDA approval in May 2020 for malaria treatment in pediatric and adult populations [7]. Artesunate has demonstrated multifaceted antitumor properties across various malignancies. The compound exerts its antineoplastic effects through diverse molecular mechanisms, including inhibition of ribosomal RNA synthesis via suppression of Fanconi anemia complementation group A (FANCA) nuclear translocation in ovarian cancer [8]. The modulation of lipid metabolism through interaction with fatty acid-binding proteins and suppression of stearoyl-CoA desaturase (SCD) expression in lung cancer [9], and induction of ferroptosis via STAT3 inhibition in hepatocellular carcinoma (HCC) [10]. Notably, artesunate has shown efficacy in breast cancer through glutathione peroxidase 4 (GPX4) inhibition, resulting in increased intracellular reactive oxygen species (ROS) and lipid peroxidation [11]. Furthermore, artemether succinate, a related compound, demonstrates synergistic effects with AXL inhibitors in triple-negative breast cancer through ROS generation, DNA damage induction, and apoptosis promotion [12], and reverses

doxorubicin resistance through up-regulation of autophagy-related genes LC3B and ATG7 [13]. The potential synergistic antitumor effects of artesunate with neuroactive agents in breast cancer treatment have also been documented [14]. Emerging evidence highlights the potential of dihydroartemisinin derivatives as alternative cancer therapeutics. Artesunate has demonstrated significant antiproliferative effects against breast cancer cell lines MCF7 and MDA-MB-231 [15], with enhanced efficacy observed through nano-structured lipid carrier encapsulation, resulting in increased apoptosis rates. In vivo studies have shown tumor growth inhibition in MCF7 xenograft models through up-regulation of apoptosis-related proteins. However, the precise molecular mechanisms underlying artesunate's anticancer effects remain incompletely elucidated.

Recent investigations have revealed artesunate's capacity to modulate various molecular pathways, including the MALAT1/PTBP1/IFIH1 axis in sepsis [16], long non-coding RNA RP-11 in hepatic epithelial-mesenchymal transition (EMT) [17], and the MALAT1/YAP signaling pathway in C918 cells [18]. In breast cancer, the long non-coding RNA (lncRNA) TUG1 has emerged as a critical regulatory element through its interaction with miR-145-5p [19–21]. Computational analyses using RNAinter have confirmed miR-145-5 p's ability to target Homeobox A5 (HOXA5) (c = 0.1619), a transcription factor implicated in tamoxifen resistance and promotion of mesenchymal-like and stem cell properties in aggressive breast cancer phenotypes [22]. The WNT signaling pathway appears to regulate HOXA5 expression through modulation of the MYC-MIZ1 complex, suggesting complex crosstalk between these pathways in breast cancer pathogenesis [23]. Given the established inhibitory effects of artesunate on breast cancer, we hypothesize that its anticancer activity may be mediated through down-regulation of lncRNA TUG1, subsequent activation of miR-145-5p, and inhibition of the HOXA5/WNT signaling axis. Therefore, this study aims to investigate the potential role of the lncRNA TUG1/miR-145-5p/HOXA5 signaling pathway in mediating the anticancer effects of artesunate in breast cancer.

## Materials and methods

This study was reviewed by the Hebei Medical University Ethics Committee, and the requirement for ethical approval was waived. The experiment exclusively involved cell line-based research and did not include any animals or human subjects. As per institutional and international guidelines, studies that do not involve living participants or vertebrate animal models are exempt from ethical review. The research adhered to all relevant scientific and ethical standards to ensure integrity, reproducibility, and responsible conduct in biomedical experimentation.

### Cell culture

Prior to initiating experiments, the cell viability was meticulously evaluated once the cells had attained the requisite density. When the viability reached 90%, the cells were sub-cultured for subsequent experimental procedures. MCF7 cells were maintained in a DMEM medium supplemented with 10% fetal bovine serum, 1%penicillin-streptomycin, and 0.01 mg/mL insulin. HCC1395 cells are cultured in RPMI 1640 medium supplemented with 10% fetal bovine serum and 1% penicillin-streptomycin. MCF10A cells were cultured in a medium specific to MCF10A. All cell cultures were incubated in a humidified environment at 37°C with 5% $CO_2$. These cells were commercial cell lines, all of which were purchased from Procell Life Science & Technology Co., Ltd. and had undergone Short Tandem Repeat (STR) identification. For seeding cells in T25 flasks, a cell density of $2 \times 10^6$ cells/ml was used, with 4 ml of medium added to each well.

### Drug stimulation

On the following day, cell proliferation was closely monitored. When the cells in the T25 flasks reached 60% − 70% confluence, drug stimulation was initiated. The Artesunate used in the experiment was obtained from GLPBIO (Catalog No: GC10889, Batch No: 1, CAS No: 88495-63-0) and used as received without further purification. The experimental groups were divided as follows: For MCF7 cells, there was a normal control group (MCF10A served as a normal reference), a

disease control group (MCF7), and treatment groups where MCF7 cells were exposed to 30 µmol/L, 120 µmol/L, and 480 µmol/L of artesunate respectively. Similarly, for HCC1395 cells, there was a normal control (MCF10A), a disease control (HCC1395), and treatment groups treated with 30 µmol/L, 120 µmol/L, and 480 µmol/L of artesunate. Cells were harvested at 0 h, 24 h, 48 h, and 72 h to assess cell proliferation using the Cell Counting Kit - 8 (CCK8) assay. Based on the optimal time point determined from the CCK8 results, samples were then collected for quantitative Polymerase Chain Reaction (qPCR) and Western Blot analyses.

## qPCR detection, RNA extraction, and quantification

For RNA extraction, thawed samples were treated with chloroform, vigorously vortexed, and incubated at room temperature for 5 min. The mixture was centrifuged at 12,000 rpm for 15 min at 4°C. The supernatant was transferred to a tube containing pre-cooled isopropanol, mixed by inversion, and left at room temperature for 10 min. After a second centrifugation at 12,000 rpm for 10 min at 4°C, the supernatant was discarded. The pellet was washed with pre-cooled 75% ethanol and subjected to a final centrifugation. Ethanol was evaporated, and the RNA was dissolved in 12 µL of RNase-free water supplemented with 1 µL of RNA inhibitor. RNA concentration and purity were quantified using a spectrophotometer.

For qPCR, the instrument was calibrated, and RNA detection was performed. A 1 µL sample was analyzed after blank collection, and data were recorded. Reverse transcription of mRNA/lncRNA was carried out to synthesize cDNA. The qPCR machine was preheated, and reagents were added to the PCR strip, gently mixed, and briefly centrifuged to eliminate bubbles. The strip was placed in the qPCR machine, and fluorescence was selected based on the target gene. Reaction parameters were set, and upon completion, melting and amplification curves were analyzed using dedicated software. Relative quantification was performed using the $2^{-\Delta\Delta Ct}$ method (Table 1).

## Western blot detection

Protein extraction was performed by adding 700 µL of pre-prepared WB lysis buffer to each EP tube after cell lysis. Samples were incubated at 4°C with gentle rotation for 30 min to ensure complete lysis. The mixture was centrifuged at 12,000 rpm for 15 min at 4°C, and the supernatant was transferred to a new EP tube. Protein quantification was conducted using a working solution prepared by mixing reagents A and B in a 50:1 ratio. A standard curve was generated using a diluted Sigma standard (2 mg/mL). Samples were prepared by mixing 8 µL of protein lysis buffer with 2 µL of extracted

**Table 1. The primer sequence.**

| Primers | Sequences |
| --- | --- |
| **TUG1-F** | GCATCTTTGCCCACATACACC |
| **TUG1-R** | CTCAGCAATCAGGAGGCACA |
| miR-92a-3p-F | CGGTCCAGTTTTCCCAGGA |
| miR-92a-3p-RT | GTCGTATCCAGTGCAGGGTCCGAGGTATTCGCACTGGATACGACAGGGAT |
| HOXA5-F | AACTCATTTTGCGGTCGCTAT |
| HOXA5-R | TCCCTGAATTGCTCGCTCAC |
| Wnt-F | AGGAGGAGACGTGCGAGAAA |
| Wnt-R | CGAGTCCATGACTTCCAGGT |
| **β-catenin-F** | AGCTTCCAGACACGCTATCAT |
| **β-catenin-R** | CGGTACAACGAGCTGTTTCTAC |
| **Arg-1-F** | GTGGAAACTTGCATGGACAAC |
| Arg-1-R | AATCCTGGCACATCGGGAATC |
| **Fizz1-F** | TTTCCTTCACCACCACCCAG |
| **Fizz1-R** | GGATGAGAAGGAGGCAAGAGG |

protein. Each well received 200 µL of the working solution and was incubated at 37°C for 25–30 min. Absorbance was measured at 562 nm, and protein concentration was calculated. Polyacrylamide gels (10%, 12%, or 15%) were prepared based on protein size. For electrophoresis, 20 µg of protein per well was mixed with 2x loading buffer containing β-mercaptoethanol, boiled for 10 minutes, briefly centrifuged, and cooled to room temperature. Gels were run at 80V until the marker entered the stacking gel, then at 120V for the separation gel. For transfer, PVDF membranes were pre-soaked in methanol. Transfer was conducted at 60V for 3 h with ice packs to maintain a low temperature. After transfer, membranes were washed with 1X TBST and blocked with 5% milk in 1X TBST for 1 h. Primary antibodies, prepared in 3% BSA, were applied overnight at 4°C. Membranes were washed three times with 1X TBST for 10 min each, followed by incubation with secondary antibodies prepared in 5% milk for 1 h. After washing, protein bands were visualized using a developing solution.

## CCK8 assay

Cell culture medium was collected in sterile 15 mL centrifuge tubes, and cells were washed once with 3 mL of NaCl. Trypsin (1 mL) was added to each T25 flask, and cells were digested until detached. The digestion was halted by adding old culture medium. The cell suspension was homogenized by pipetting and centrifuged at 1,000 rpm for 5 min. The supernatant was discarded, and cells were resuspended in 1–3 mL of fresh culture medium. Cells were counted using Trypan Blue and seeded into a 96-well plate at a density of 2,000 cells per well in 200 µL of medium. Four replicates were prepared for each cell type. For the CCK8 proliferation assay, the CCK8 working solution (10 µL of CCK8 stock solution in 100 µL of serum-free medium) was prepared. The old medium was replaced with the working solution (110 µL per well), and plates were incubated at 37°C for 1.5 h. After incubation, 95 µL of the solution was transferred to a new 96-well plate, and absorbance was measured at 450 nm using a microplate reader.

## Plasmid transfection and drug stimulation

Plasmid transfection was initiated when cells reached 60–70% confluency. siRNA and jetPRIME buffer were combined, vortexed for 10 seconds, and incubated for 5 minutes. jetPRIME reagent was added, vortexed for 5 seconds, and incubated at room temperature for 10 minutes. The mixture, along with OPTI-MEM medium, was added to the culture dish and incubated. After 6 hours, the medium was replaced with the original cell culture medium. Forty-eight hours post-transfection, drug stimulation with artesunate was performed. Cells were collected 48 hours later for further analysis.

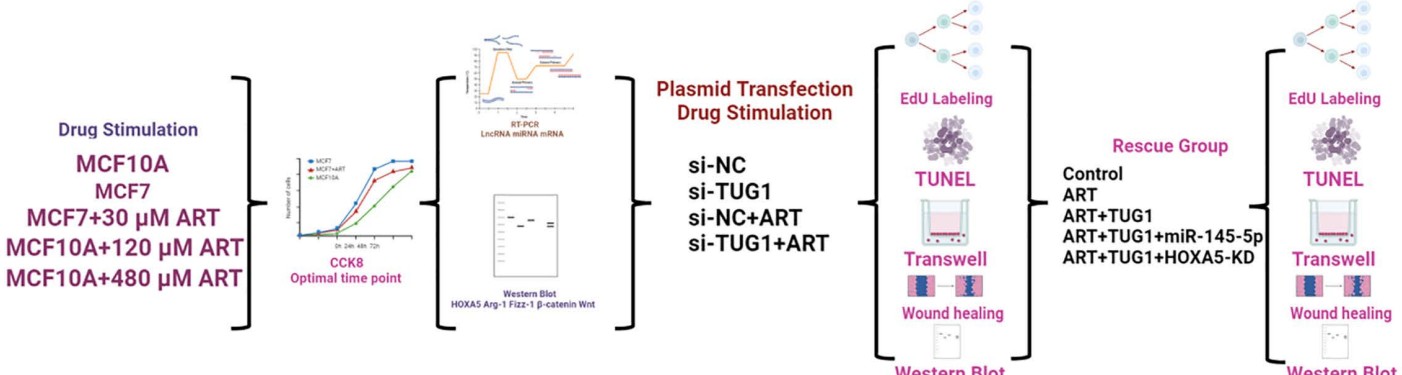

## EdU labeling

Cells were cultured in a 6-well plate, with coverslips added if necessary. After overnight incubation, drug treatment or other stimuli were applied. A 2X EdU working solution (20 µM) was prepared by diluting EdU (10 mM) in culture medium at a

1:500 ratio. An equal volume of pre-warmed 2X EdU solution was added to each well to achieve a final 1X EdU concentration. Cells were incubated for 2h. After labeling, the medium was removed, and cells were fixed with 1 mL of fixation solution for 15 min at room temperature. For flow cytometry, adherent cells were detached using trypsin, resuspended in medium, and fixed. Cells were washed three times with 1 mL of wash buffer (3–5 min each), permeabilized with 1 mL of permeabilization solution for 10–15 min, and washed again. For the Click reaction, the Click Additive Solution was prepared by dissolving one vial of Click Additive C0078S in 1.3 mL of deionized water. The reaction system (500 µL per well) was set up, and specific procedures were followed as per the protocol.

## TUNEL

Cells were initially rinsed with PBS or HBSS, followed by fixation using either immunostaining fixative or 4% paraformaldehyde for 30 min. After fixation, cells were washed again with PBS or HBSS. A strong permeabilization solution or PBS containing 0.3% Triton X-100 was applied, and cells were incubated at room temperature for 5 min. Subsequently, the samples were washed twice with PBS or HBSS, and 50 µl of TUNEL detection solution was added, followed by incubation at 37°C in the dark for 60 minutes. Post-incubation, the samples were washed three times with PBS or HBSS. Finally, the samples were mounted using an anti-fade mounting medium and observed under a fluorescence microscope. Cy3 exhibited an excitation wavelength of 550 nm and an emission wavelength of 570 nm, producing red fluorescence.

## Transwell invasion

Matrigel was thawed overnight on ice at 4°C. The following morning, it was diluted to a final concentration of 1 mg/ml using a serum-free medium pre-chilled at 4°C, with all steps performed on ice. Between 100–200 µl of the diluted Matrigel was added to the center of the bottom of the upper chamber and allowed to solidify into a gel by incubating at 37°C for 4–5 h. The 24-well plate was then removed from the incubator, and 500 µl of serum was added to the lower chamber. Subsequently, 200 µl of cells with adjusted density were gently added along the inner wall of the upper chamber. The plate was returned to the incubator, and samples were collected at 16h, 24h, and 48h intervals. All pipette tips and experimental dishes that came into contact with Matrigel were pre-cooled at −20°C overnight. The cell names were labeled on the walls and caps of 15 ml centrifuge tubes. The cell culture medium was collected in sterile 15 ml centrifuge tubes and washed with 3 ml of NaCl solution 1–2 times. After washing, 1 ml of trypsin was added to each T25 flask and placed in the incubator. The cells were digested until they detached and assumed a sandy slip appearance, then old culture medium was added to halt the digestion. The mixture was pipetted up and down to mix well, transferred to 15 ml centrifuge tubes, and centrifuged at 1,000 rpm for 5 min. The supernatant was discarded, and 1–3 ml of culture medium was added to resuspend the cells. Trypan Blue was used at a 1:1 ratio to adjust the cell density. Transwell invasion chamber was removed from the incubator and transferred to a new 24-well plate. Then, 500 µl of methanol was added per well to the empty lower chamber and fixed for 25–30 min. The fixed chamber was transferred to a new 24-well plate containing 500 µl per well of 0.1% crystal violet solution and stained for 25–30 min. The stained chamber was rinsed twice in double-distilled water, excess cells were scraped off from the inside of the chamber with a cotton swab, the chamber was inverted onto absorbent paper to air dry, and the outside of the chamber was marked. Photographs were taken under a microscope at 10X magnification.

## Scratch migration

The cell names were labeled on the walls and caps of 15 ml centrifuge tubes. The cell culture medium was collected in sterile 15 ml centrifuge tubes and washed with 3 ml of NaCl solution 1–2 times. After washing, 1 ml of trypsin was added to each T25 flask and placed in the incubator. The cells were digested until they detached and appeared as a sandy slip, and then an old culture medium was added to stop the digestion. The mixture was pipetted up and down to mix well, transferred to 15 ml centrifuge tubes, and centrifuged at 1,000 rpm for 5 min. The supernatant was discarded, and 1–3 ml of culture medium was added to resuspend the cells. Trypan Blue was used at a 1:1 ratio to adjust the cell density. Then,

2 ml of cell suspension was added per well in a 6-well plate, which was then placed in the incubator, with the medium being changed every 2–3 days. The following day, the old culture medium was removed using a 1 ml pipette, and a 200 µl yellow pipette tip was employed to draw a straight line from top to bottom in the middle of each well. Subsequently, 2 ml of culture medium was gently added along the well wall, and microscope images (5X objective) were immediately taken to record migration at 0h. The medium change steps were repeated as described above, and migration was recorded at 24h through additional microscope images.

### Dual-luciferase reporter gene

The reporter gene cell lysis solution was meticulously mixed and added to the cells as follows to ensure complete lysis. For adherent cells, the cell culture medium was removed, and an appropriate volume of the reporter gene cell lysis solution was added according to the table below. The lysis solution can be stored at 4°C for up to one month; however, it is recommended to prepare a fresh batch according to the experimental requirements before use. The culture plate was placed on a shaker and gently agitated to ensure the lysis solution completely covered the cell monolayer. The culture plate was then shaken at room temperature for 15 min. After complete lysis, the cells were centrifuged at 10,000–15,000 rpm for 3–5 min, and the supernatant was collected for analysis. It was noted that the fluorescent enzyme could be measured immediately after cell lysis or frozen for later analysis. Samples could be stored at −20°C for up to one month; however, for extended storage, it is recommended to store them at −80°C. Thawed samples should be fully dissolved and brought to room temperature before analysis. Repeated freeze-thaw cycles were discouraged as 2–3 cycles might gradually result in the loss of luciferase reporter enzyme activity. The luciferase assay reagent was thawed and brought to room temperature. The instrument operating manual was followed to initiate the chemiluminescence analyzer or multifunctional microplate reader with chemiluminescence detection capability. The measurement interval was set to 2 seconds and the measurement duration to 10 seconds. For each sample measurement, 20–100 µl of the sample was taken (100 µl was used if the sample volume was sufficient; if the sample volume was insufficient, the amount could be adjusted accordingly, but consistency across the same batch was crucial). Subsequently, 100 µl of luciferase assay reagent was added and thoroughly mixed with a pipette, and the relative light unit (RLU) was measured. Upon completing the firefly luciferase assay, an equal volume of Renilla luciferase assay working solution was added to measure the RLU. The ratio of the two luciferases (reporter gene: internal control gene) was calculated, and the differences in ratios between different groups were compared.

### Statistical analysis

Statistical analyses were conducted using GraphPad 9.0. Data are presented as mean ± standard deviation (S.D.). Statistical significance was determined using an unpaired two-tailed Student's t-test. A p-value of less than 0.05 was considered statistically significant.

## Results

### 1. Artesunate suppresses the proliferation of MCF7 tumor cells and inhibits the WNT/β-catenin pathway via the lncRNA TUG1/miR-145-5p/HOXA5 axis

To evaluate the cytotoxic effects of Artesunate on breast cancer cells, the half-maximal inhibitory concentration (IC50) was determined for MCF7 and HCC1395 breast cancer cell lines, as well as for normal breast epithelial cells (MCF-10A). Using CCK8 assay, cells were exposed to varying concentrations of Artesunate, and viability was assessed. The IC50 values were calculated as 690 µM for MCF7, 434 µM for HCC1395, and 861 µM for MCF-10A, indicating a more pronounced inhibitory effect on cancer cells compared to normal cells. Further viability assessments were conducted at Artesunate concentrations of 30 µM, 120 µM, and 480 µM. Notably, MCF7 and HCC1395 cells exhibited significantly reduced viability

with increasing Artesunate concentrations, whereas MCF-10A cells remained relatively unaffected. Based on these findings, 120 μM was selected as the standard treatment concentration for subsequent experiments (Fig 1A).

To elucidate the molecular mechanisms underlying Artesunate's effects, potential binding interactions between lncRNA TUG1, miR-145-5p, and HOXA5 were investigated. Using bioinformatics databases (starBASE and micRDB), binding sequences were predicted and cloned into dual-luciferase reporter vectors, alongside mutant constructs. Co-transfection of 293T cells with miR-145-5p mimics and respective plasmids revealed that miR-145-5p significantly suppressed fluorescence expression when bound to lncRNA TUG1-WT and HOXA5–3'-UTR-WT but not their mutant counterparts. These results confirm specific binding interactions between miR-145-5p and both lncRNA TUG1 and HOXA5 (Fig 1B).

To assess the transcriptional impact of Artesunate, MCF7 and MCF-10A cells were treated with varying concentrations of Artesunate (30 μM, 120 μM, and 480 μM), and RNA expression levels were quantified via RT-qPCR. In MCF7 cells, Artesunate induced a dose-dependent reduction in lncRNA TUG1, HOXA5, WNT, β-catenin, Fizz1, and Arg-1 expression, while miR-145-5p levels increased significantly. In contrast, MCF-10A cells exhibited minimal changes in gene expression, further highlighting the selective action of Artesunate on cancer cells (Fig 1C).

To assess the transcriptional impact of Artesunate, MCF-10A and HCC1395 cells were treated with varying concentrations of Artesunate (30 μM, 120 μM, and 480 μM), and protein expression levels were quantified via Western blot. Results showed the effects of Artesunate on protein expression in MCF7, HCC1395, and MCF-10A cells. Treatment with Artesunate resulted in a dose-dependent decrease in HOXA5, WNT, β-catenin, Fizz1, and Arg-1 protein levels in both MCF7 and HCC1395 cells, while MCF-10A cells showed no significant alterations. These findings suggest that Artesunate selectively inhibits key signaling pathways in breast cancer cells (Fig 2A,B).

## 2. Artesunate regulates the proliferation and apoptosis of breast cancer cells through lncRNA TUG1

Following standard culture protocols, cells were expanded to logarithmic growth phase, digested, and counted prior to seeding in 6-well plates. Transfection was performed using lipid-mediated delivery of siRNA targeting lncRNA TUG1 (siRNA lncRNA TUG1) alongside negative controls (siNC). After 24-hour transfection, cells were processed and transferred to 96-well plates for adherence. Initial proliferation activity was assessed at baseline (0 hours), followed by Artesunate treatment at 24, 48, and 72-hour intervals. Cell viability was quantified using CCK8 assay, revealing that lncRNA TUG1 knockdown significantly inhibited proliferation, with Artesunate treatment further enhancing this suppression (Fig 3A and B).

For proliferation assessment, cells were cultured and transfected as described above, then seeded in 24-well plates. Following 48-hour transfection, Artesunate was administered, and proliferation was evaluated using EdU assay. Results demonstrated reduced red fluorescence intensity, indicating that lncRNA TUG1 knockdown coupled with Artesunate treatment significantly impaired cellular proliferation (Fig 3C and D).

Using similar culture and transfection protocols, cells were treated with Artesunate post-transfection for apoptosis assessment via TUNEL assay. Increased red fluorescence intensity was observed, demonstrating enhanced apoptosis following lncRNA TUG1 knockdown, with Artesunate treatment further augmenting this effect (Fig 3E and F).

## 3. Artesunate inhibits M2 polarization and suppresses the migratory and invasive capacities of breast cancer cells via the lncRNA TUG1-mediated WNT/β-catenin signaling pathway

MCF7 and HCC1395 cells were cultured and expanded to logarithmic growth. After digestion and counting, cells were seeded into 6-well plates. Lipid-mediated transfection of siRNA targeting lncRNA TUG1 (siRNA lncRNA TUG1) and negative control (siNC) was performed. After 24 hours of transfection, cells were re-digested, counted, and seeded into Transwell chambers, followed by Artesunate treatment. Invasion capacity was evaluated using Transwell chambers, where lncRNA TUG1 knockdown reduced invasive potential, with Artesunate treatment further suppressing this activity (Fig 4A and B).

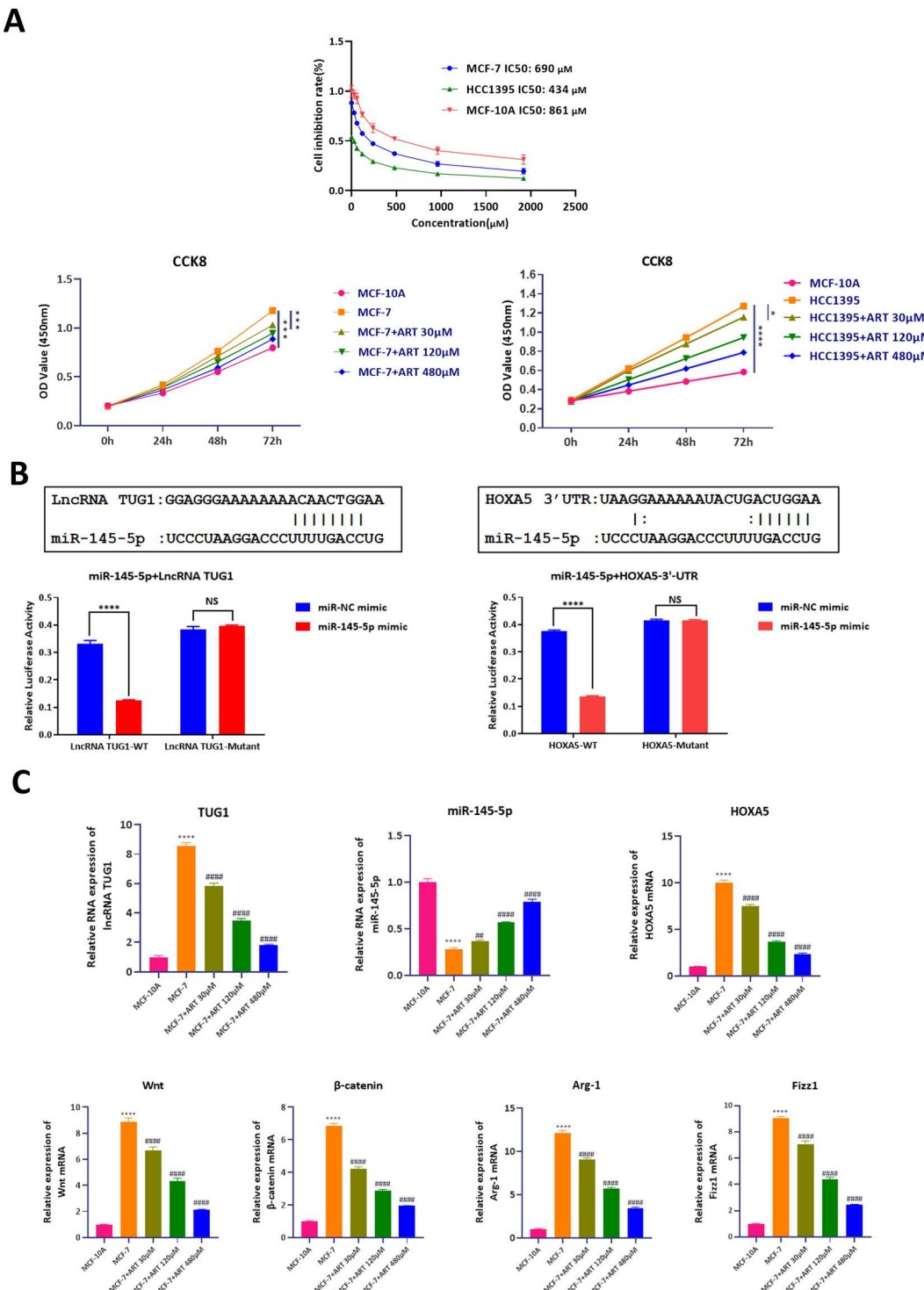

**Fig 1. Inhibitory effects of different concentrations of artesunate on MCF7 cells and HCC1395 cells on cell viability, LncRNA, miR-145-5p, HOXA5 and their pathways.** A. The half-maximal inhibitory concentration (IC50) of artesunate on MCF7, HCC1395, and MCF10A cells was determined by the CCK8 assay. The viability of MCF7 cells and HCC1395 was assessed following treatment with artesunate at concentrations of 30 μM, 120 μM, and 480 μM. Two-way ANOVA analysis, N = 4. B. Binding interactions between lncRNA TUG1 and miR-145-5p, as well as between HOXA5 and miR-145-5p, were identified using the dual-luciferase reporter assay. C. Expression profiles of lncRNA TUG1, miR-145-5p, HOXA5, WNT, β-catenin, Fizz1, and Arg-1 RNA in MCF10A and MCF7 cells under varying concentrations of artesunate. One-way ANOVA analysis, N = 3. **P < 0.01, ***P < 0.001, ****P < 0.0001.

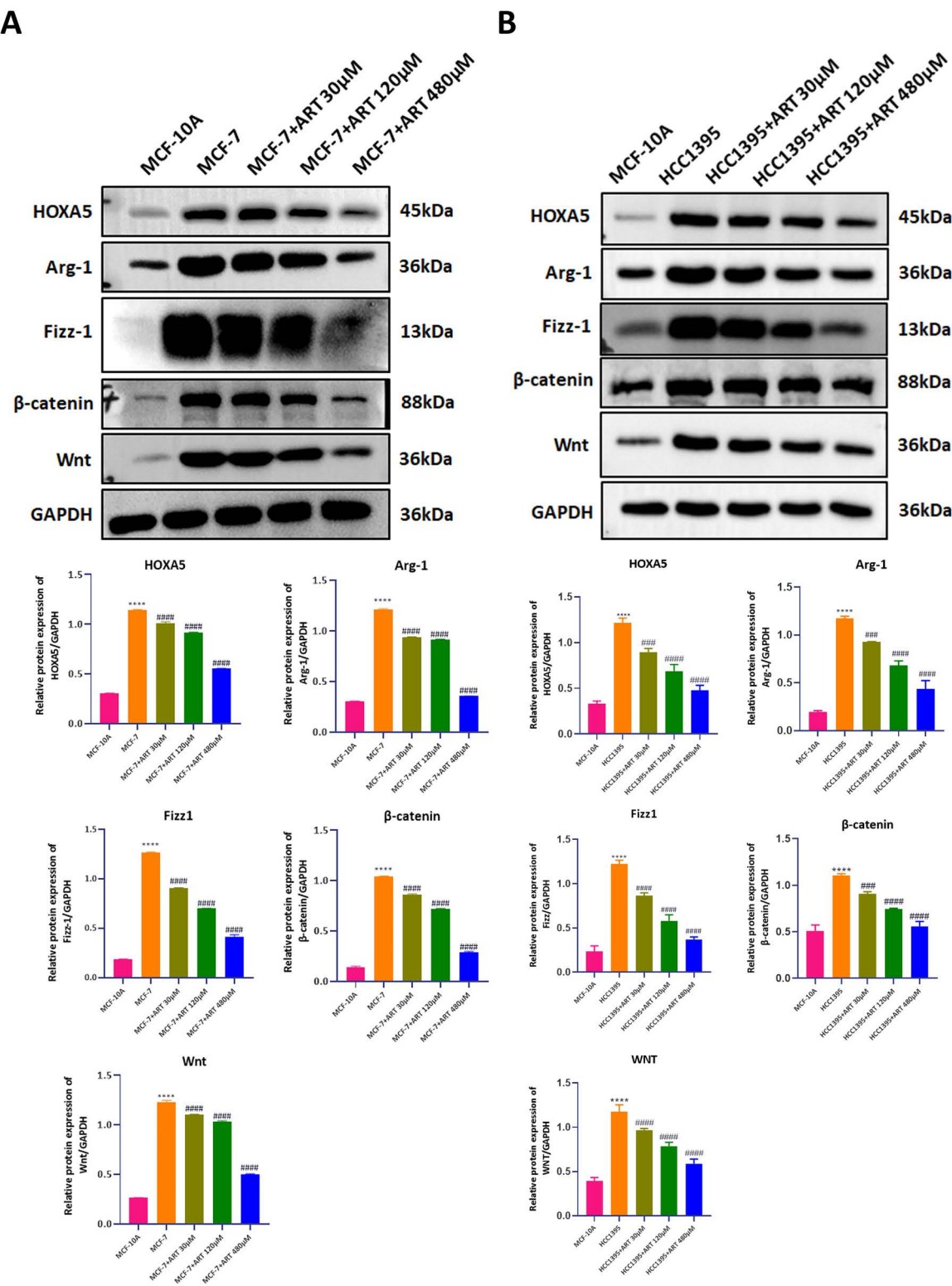

**Fig 2. Effects of different concentrations of artesunate on pathway proteins in MCF7 cells and HCC1395 cells.** A&B. Western blot analysis of protein expression levels of HOXA5, WNT, β-catenin, Fizz1, and Arg-1 in MCF-7 cells and HCC 1395 cells treated with different concentrations of artesunate, MCF10A was employed as a control. One-way ANOVA analysis, N = 3. **P < 0.01, ***P < 0.001, ****P < 0.0001.

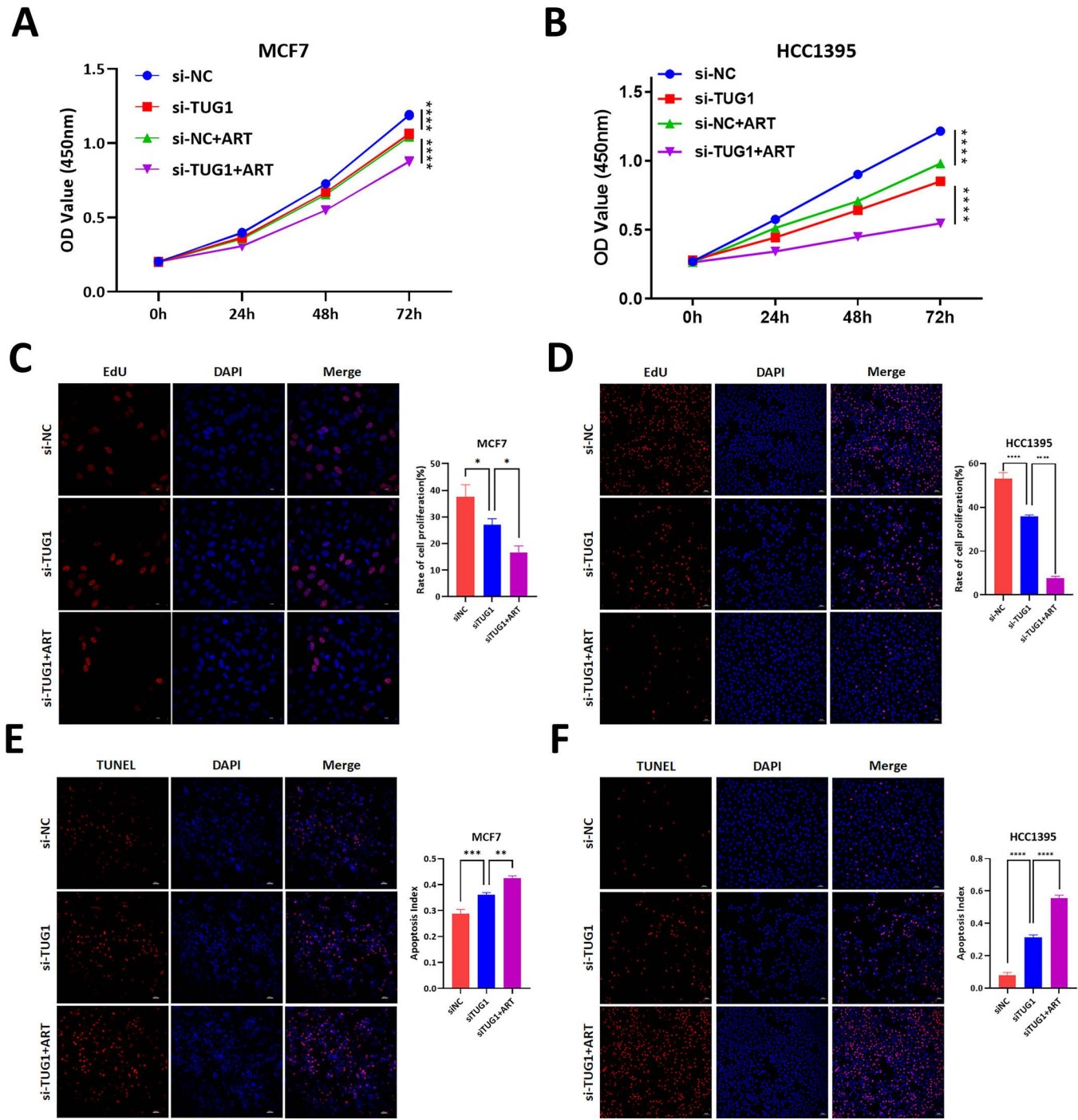

**Fig 3. Effects of lncRNA TUG1 knockdown and stimulation with Artesunate on the proliferation and apoptosis of MCF7 and HCC1395 cells.** A and B. CCK8 assay was used to measure OD values at 450 nm of MCF-7 and HCC1395 cells at 24 hours, 48 hours, and 72 hours after either knockdown of long-non-coding RNA TUG1 alone or in combination with artesunate treatment. Two-way ANOVA analysis, N = 4. C and D. EdU assay was used to detect the cell proliferation rates of MCF-7 and HCC1395 cells after either knockdown of long-non-coding RNA TUG1 alone or in combination with artesunate treatment. One-way ANOVA analysis, N = 3. E and F. TUNEL assay was used to measure the proportion of apoptotic cells in MCF-7 and HCC1395 cells after either knockdown of long-non-coding RNA TUG1 alone or in combination with artesunate treatment. One-way ANOVA analysis, N = 3. **P < 0.01, ***P < 0.001, ****P < 0.0001.

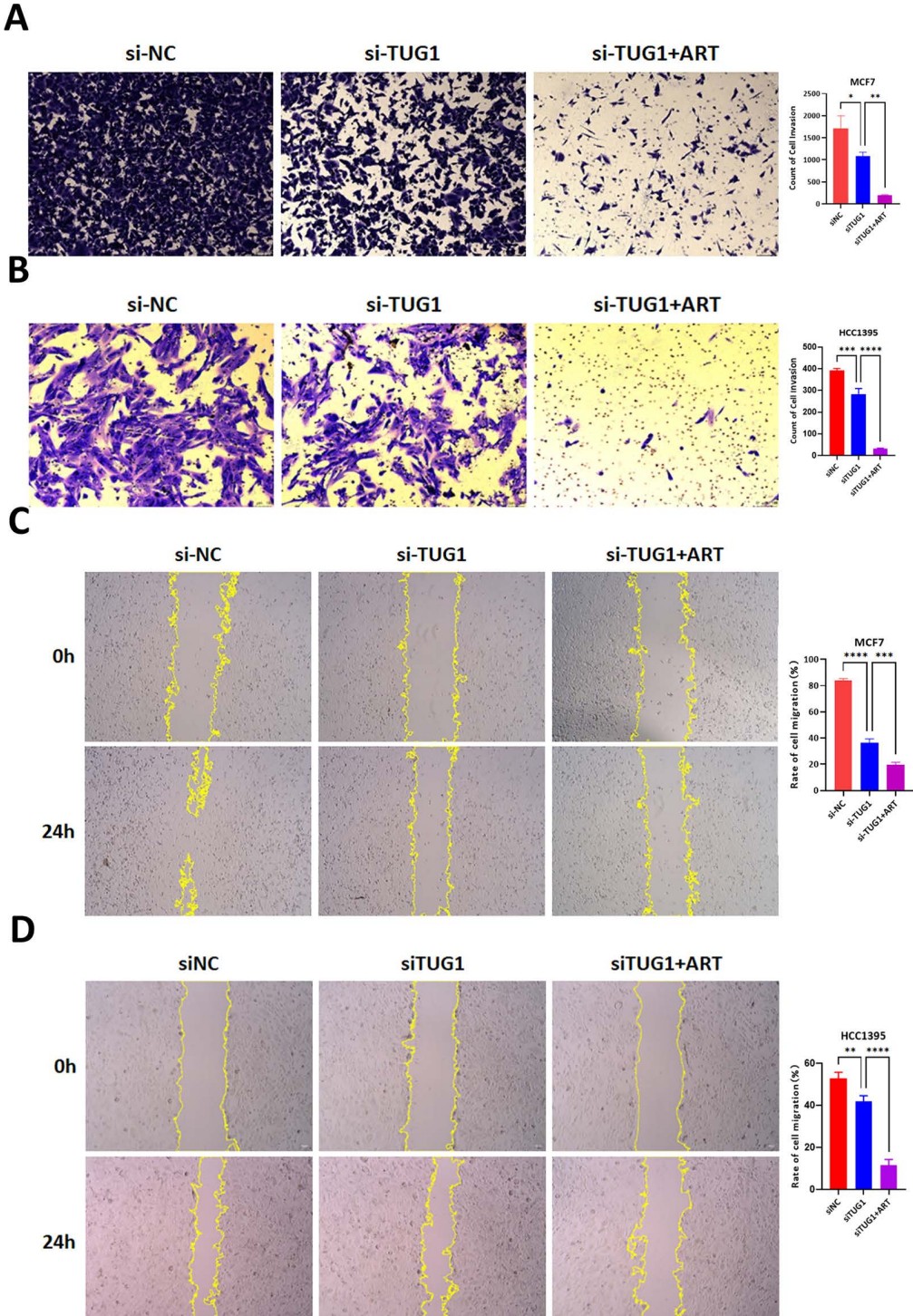

**Fig 4. Effects of lncRNA TUG1 knockdown and stimulation with Artesunate on the migration, invasion of MCF7 and HCC1395 cells.** A and B. Transwell assay was used to detect the number of invasive cells of MCF-7 and HCC1395 cells after either knockdown of long non-coding RNA TUG1 alone or in combination with artesunate treatment. C and D. Wound healing assay was used to detect the proportion of migrating cells of MCF-7 and HCC1395 cells after either knockdown of long non-coding RNA TUG1 alone or in combination with artesunate treatment. One-way ANOVA analysis, N=3. **P<0.01, ***P<0.001, ****P<0.0001.

MCF7 and HCC1395 cells were cultured and expanded to logarithmic growth. After digestion and counting, cells were seeded into 6-well plates. Lipid-mediated transfection of siRNA targeting lncRNA TUG1 (siRNA lncRNA TUG1) and negative control (siNC) was performed. After 24 hours of transfection, a vertical scratch was made at the center of each well using a pipette tip to create a wound. The width of the scratch was recorded at that time. After 24 hours of further culture, the scratch results were observed. For migration analysis, transfected cells were seeded in 6-well plates and subjected to scratch assay. Wound closure was monitored, showing that lncRNA TUG1 knockdown impaired migration, with Artesunate treatment further inhibiting this process (Fig 4C and D).

Under the MCF7 and HCC1395 cells cell culture conditions, cells were digested and counted during logarithmic growth, then seeded into 6-well plates. Subsequently, lipid-mediated transfection of siRNA targeting lncRNA TUG1 (siRNA lncRNA TUG1) and negative control (siNC) was performed. After 72 hours of transfection, cell extracts were collected for protein analysis via Western blot to assess the expression levels of HOXA5, WNT, β-catenin, Fizz1, and Arg-1. Knockdown of lncRNA TUG1 inhibited the protein expression of HOXA5, WNT, β-catenin, Fizz1, and Arg-1. Additionally, after Artesunate treatment, protein expression levels were further suppressed (Fig 5A and B).

## 4. Artesunate inhibits the malignant phenotype of MCF7 cells by modulating the miR-145-5p and HOXA5 molecules mediated by lncRNA TUG1

To validate the role of lncRNA TUG1 through miR-145-5p and HOXA5, we investigated the behavior of Artesunate-regulated lncRNA TUG1 using miR-145-5p and HOXA5.RT-PCR analysis revealed distinct expression patterns across experimental groups. Compared to the control (Ctrl) group, the ART group exhibited a significant reduction in lncRNA TUG1 expression, accompanied by an upregulation of miR-145-5p and downregulation of HOXA5. In the ART + TUG1-OE group, overexpression of lncRNA TUG1 resulted in elevated TUG1 and HOXA5 levels, while miR-145-5p expression was suppressed. Notably, the ART + TUG1-OE + miR-145-5p mimic group demonstrated no significant change in TUG1 expression but showed a marked increase in miR-145-5p and a decrease in HOXA5 compared to the ART + TUG1-OE + NC mimic group. Similarly, the ART + TUG1-OE + siHOXA5 group exhibited no significant alterations in TUG1 or miR-145-5p expression but displayed a significant reduction in HOXA5 levels compared to the ART + TUG1-OE + siNC group (Fig 6A).

CCK8 assays demonstrated that ART treatment significantly reduced the OD value of MCF7 cells at 72 hours compared to the Ctrl group. Overexpression of TUG1 in the ART + TUG1-OE group reversed this effect, leading to increased cell proliferation. However, the introduction of miR-145-5p mimic in the ART + TUG1-OE + miR-145-5p mimic group significantly decreased the OD value, while the ART + TUG1-OE + siHOXA5 group showed a modest reduction (Fig 6B).

EdU assays corroborated the CCK8 findings, showing a significant reduction in proliferating MCF7 cells in the ART group compared to the Ctrl group. TUG1 overexpression in the ART + TUG1-OE group restored proliferation levels, which were subsequently diminished by miR-145-5p mimic in the ART + TUG1-OE + miR-145-5p mimic group. The ART + TUG1-OE + siHOXA5 group exhibited a slight decrease in proliferating cells (Fig 6C).

Scratch assays indicated that, compared to the Ctrl group, the migration distance of MCF7 cells in the ART group was significantly reduced. Scratch assays revealed that ART treatment significantly impaired the migration of MCF7 cells compared to the Ctrl group. TUG1 overexpression in the ART + TUG1-OE group enhanced migration, which was significantly reduced by miR-145-5p mimic in the ART + TUG1-OE + miR-145-5p mimic group. The ART + TUG1-OE + siHOXA5 group showed a modest reduction in migration distance (Fig 6D).

Transwell invasion assays demonstrated that, compared to the Ctrl group, the number of invading MCF7 cells in the ART group was significantly reduced. Transwell invasion assays demonstrated that ART treatment significantly reduced the number of invading MCF7 cells compared to the Ctrl group. TUG1 overexpression in the ART + TUG1-OE group increased invasion, though not to the level of the Ctrl group. The introduction of miR-145-5p mimic in the ART + TUG1-OE + miR-145-5p mimic group significantly decreased invasion, while the ART + TUG1-OE + siHOXA5 group showed a reduction in invasive capacity (Fig 6E).

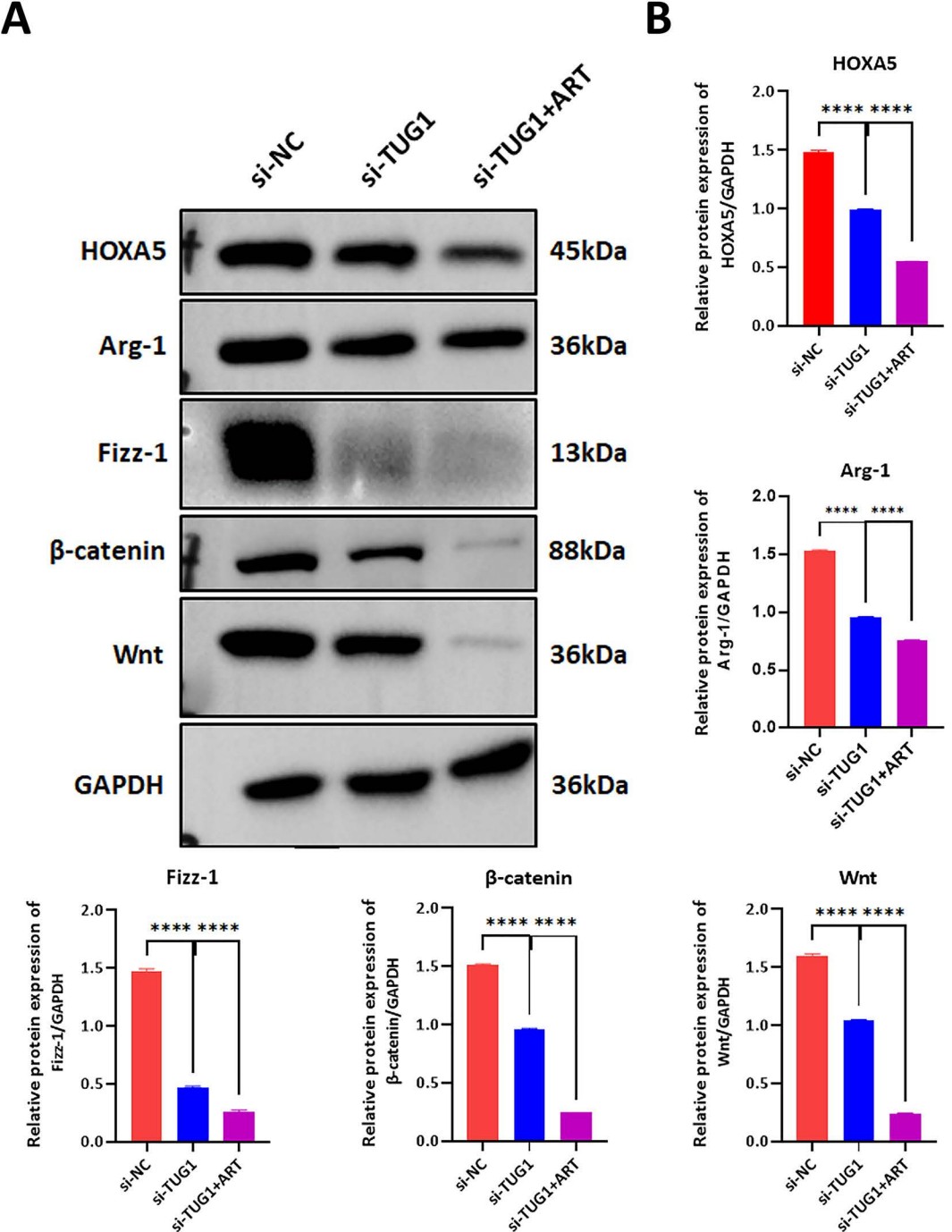

**Fig 5. Effects of lncRNA TUG1 knockdown and stimulation with Artesunate on the WNT/β-catenin signaling pathway in MCF7.** A and B: Western Blot assay was used to detect the protein expression levels of HOXA5, WNT, β-catenin, Fizz1, and Arg-1 in MCF-7 and HCC1395 cells after either knockdown of long non-coding RNA TUG1 alone or in combination with artesunate treatment. Si-NC was used as a control. One-way ANOVA analysis, N = 3. **P < 0.01, ***P < 0.001, ****P < 0.0001.

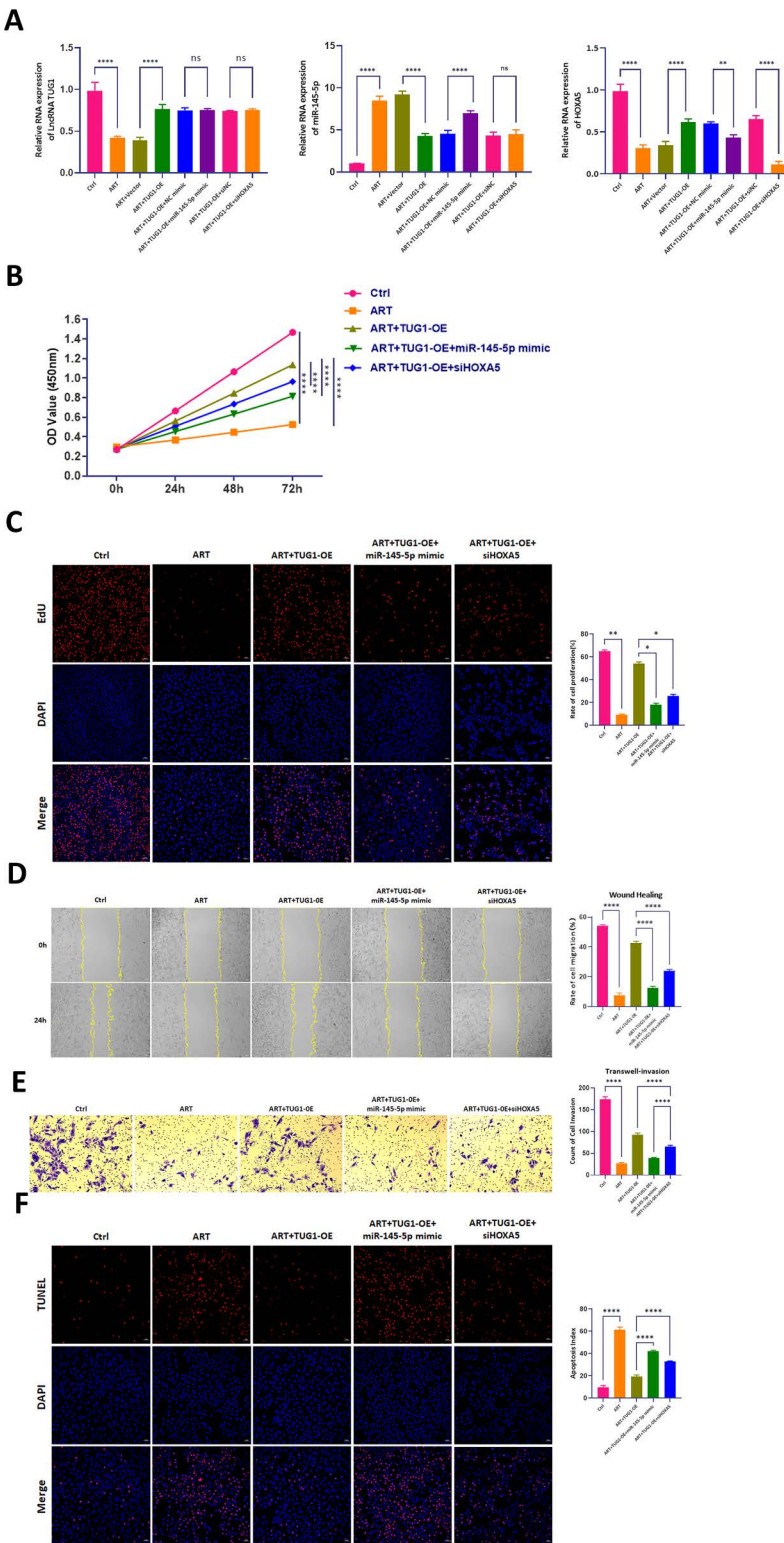

**Fig 6. Rescue experiments on the cellular phenotypes of ATR-treated MCF7 cells by miR-145-5p and HOXA5.** A. RT-PCR was used to detect the RNA expression levels of TUG1, miR-145-5p, and HOXA5 in the Ctrl group, ATR group, ATR+Vector group, ATR+TUG1-OE group, ATR+TUG1-OE+NC mimic group, ATR+TUG1-OE+miR-145-5p mimic group, ATR+TUG1-OE+si-NC group, and ATR+TUG1-OE+si-HOXA5

group. One-way ANOVA analysis, N = 3. **P < 0.01, ***P < 0.001, ****P < 0.0001. B. CCK8 assay was used to detect the OD values at 450 nm of the Ctrl group, ATR group, ATR + TUG1-OE group, ATR + TUG1-OE + miR-145-5p mimic group,and ATR + TUG1-OE + si-HOXA5 group at 24 hours, 48 hours, and 72 hours. Two-way ANOVA analysis, N = 3. **P < 0.01, ***P < 0.001, ****P < 0.0001. C. EdU assay was used to detect cell proliferation rate of the Ctrl group, ATR group, ATR + TUG1-OE group, ATR + TUG1-OE + miR-145-5p mimic group,and ATR + TUG1-OE + si-HOXA5 group. One-way ANOVA analysis, N = 3. **P < 0.01, ***P < 0.001, ****P < 0.0001. D. Wound healing assay was used to detect cell migration distance of the Ctrl group, ATR group, ATR + TUG1-OE group, ATR + TUG1-OE + miR-145-5p mimic group, and ATR + TUG1-OE + si-HOXA5 group. One-way ANOVA analysis, N = 3. **P < 0.01, ***P < 0.001, ****P < 0.0001. E. Transwell assay was used to detect the number of invasive cells of the Ctrl group, ATR group, ATR+Vector group, ATR + TUG1-OE group, ATR + TUG1-OE + miR-145-5p mimic group,and ATR + TUG1-OE + si-HOXA5 group. One-way ANOVA analysis, N = 3. **P < 0.01, ***P < 0.001, ****P < 0.0001. F. TUNEL assay was used to detect the number of apoptotic cells of the Ctrl group, ATR group, ATR + TUG1-OE group, ATR + TUG1-OE + NC mimic group, ATR + TUG1-OE + miR-145-5p mimic group,and ATR + TUG1-OE + si-HOXA5 group. One-way ANOVA analysis, N = 3. **P < 0.01, ***P < 0.001, ****P < 0.0001.

TUNEL assays indicated that ART treatment significantly increased the proportion of apoptotic MCF7 cells compared to the Ctrl group. TUG1 overexpression in the ART + TUG1-OE group reduced apoptosis, though not below Ctrl levels. The ART + TUG1-OE + miR-145-5p mimic group exhibited a significant increase in apoptosis, while the ART + TUG1-OE + siHOXA5 group also showed elevated apoptotic rates (Fig 6F).

Western blot analysis demonstrated that, compared to the Ctrl group, the protein expression levels of WNT, β-catenin, Fizz1, and Arg-1 in MCF7 cells of the ART group were significantly reduced. Compared to the ART group, the ART + TUG1-OE group showed a significant increase in the protein expression levels of WNT, β-catenin, Fizz1, and Arg-1 in MCF7 cells. Compared to the ART + TUG1-OE group, the ART + TUG1-OE + miR-145-5p mimic group exhibited a significant decrease in the protein expression levels of WNT, β-catenin, Fizz1, and Arg-1 in MCF7 cells. Compared to the ART + TUG1-OE group, the ART + TUG1-OE + siHOXA5 group showed a slight decrease in the protein expression levels of WNT, β-catenin, Fizz1, and Arg-1 (Fig 7).

## Discussion

The experimental results demonstrated that the inhibitory effect of Artesunate on breast cancer cells was amplified with increasing concentrations. Artesunate, at varying concentrations, also suppressed the protein expression of HOXA5, WNT, β-catenin, Fizz1, and Arg-1, thereby modulating the WNT-β-catenin signaling pathway and ultimately curbing the proliferation of breast tumor cells. Moreover, Artesunate at different concentrations inhibited the transcriptional expression of lncRNA TUG1, which in turn promoted the transcription of miR-145-5p and suppressed the transcription of HOXA5, WNT, β-catenin, Fizz1, and Arg-1. These findings suggest that Artesunate exerts its inhibitory effects on breast cancer, likely through modulation of the WNT/β-catenin signaling pathway by altering TUG1 expression.

### Examples of natural products with cancer-suppressive properties

Artesunate is not the sole naturally derived molecule shown to have inhibitory effects on cancer cells. Curcumin, for instance, a compound extracted from the turmeric plant (Curcuma longa), has demonstrated in vitro anti-cancer activities, including anti-proliferative, pro-apoptotic, anti-metastatic, and anti-angiogenic effects. Additionally, curcumin regulates autophagy, reverses multidrug resistance, modulates immunity, and enhances chemotherapy efficacy [23,24]. Artesunate induces caspase-dependent apoptosis in breast cancer cells by inhibiting HSP70 protein. In this study, the anti-apoptotic effect of Artesunate is regulated through the lncRNA/miRNA/mRNA cascade reaction, which modulates HOXA5 and subsequently affects the Wnt/β-catenin signaling pathway. Further research is needed to determine which specific caspase protein is involved [25]. Artesunate can promote the degradation of hexokinase 1 (HK1) by targeting it, while simultaneously downregulating the levels of HIF1α and PKM2, thereby inhibiting aerobic glycolysis in esophageal cancer progression. The exacerbation of aerobic glycolysis is a novel metabolic indicator of cancer progression. Whether HOXA5 promotes breast cancer progression by enhancing glycolysis suggests that, based on molecular docking

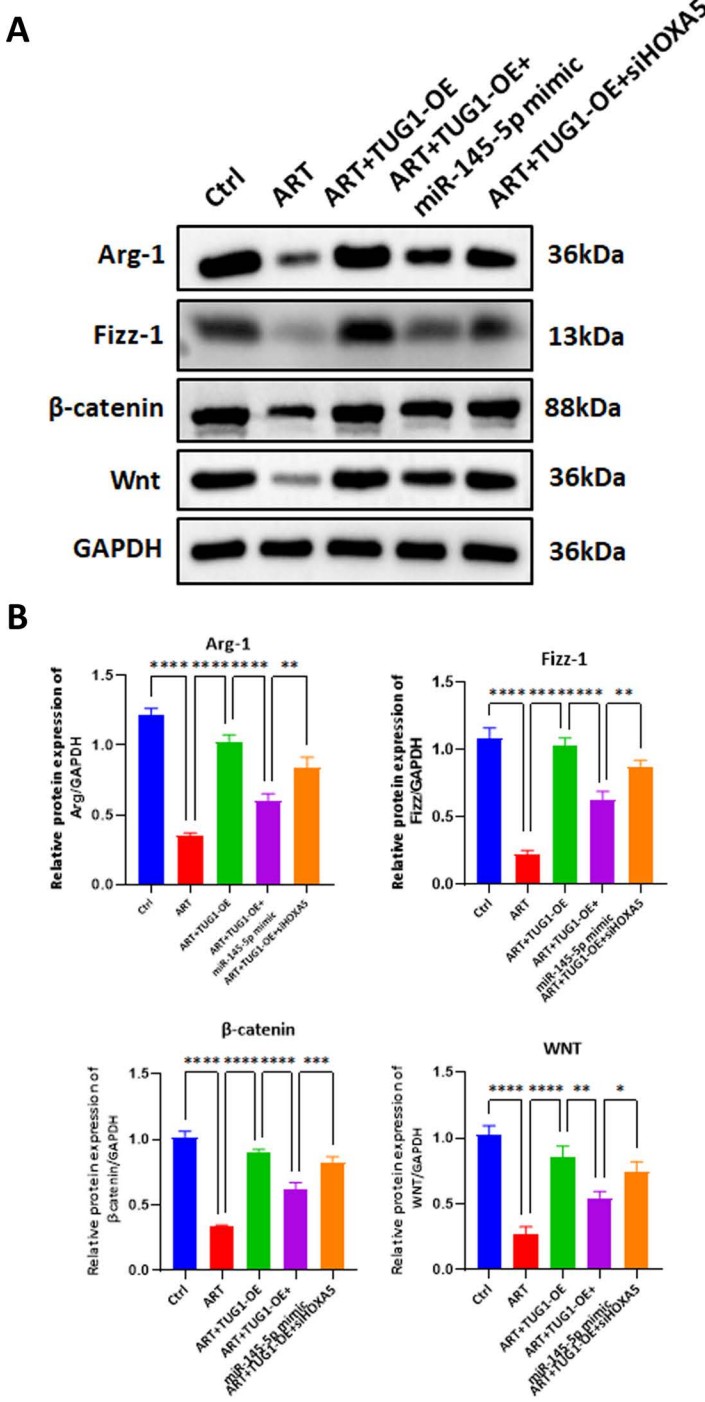

**Fig 7. Rescue experiments on the signaling pathway of ATR-treated MCF7 cells by miR-145-5p and HOXA5.** A&B.Western Blot experiments showed the protein expression levels of WNT, β-catenin, Fizz1, and Arg-1 in the Ctrl group, ATR group, ATR+Vector group, ATR+TUG1-OE group, ATR+TUG1-OE+miR-145-5p mimic group,and ATR+TUG1-OE+si-HOXA5 group. One-way ANOVA analysis, N=3. **P<0.01, ***P<0.001, ****P<0.0001.

experiments, Artesunate could inhibit the aggravation of malignant phenotypes through lncRNA TUG1-mediated glycolysis suppression [26]. Artesunate inhibits colorectal cancer progression by suppressing reactive oxygen species-induced cellular senescence and autophagy. Whether its effect on breast cancer is mediated through the lncRNA/miRNA/mRNA signaling pathway, influencing the expression of senescence-related proteins and exacerbating endoplasmic reticulum stress, remains unclear [27]. Artesunate can reverse paclitaxel resistance in cancer cells by enhancing lysosomal function. It can also modulate the neurite outgrowth inhibitor protein B receptor to regulate sorafenib resistance in hepatocellular carcinoma. Research on breast cancer drug resistance and the targets mentioned in this study provides new insights [28,29]. Artesunate induces a novel form of cell death—ferroptosis—in ovarian serous carcinoma, rather than solely relying on apoptosis [30]. Artesunate inhibits melanoma progression by suppressing the STAT3 signaling pathway. In this study, it targets the long non-coding RNA TUG1 to inhibit ovarian cancer growth. The development of comprehensive targets for Artesunate can serve as a new summary [8,31]. New derivatives of Artesunate can exert cytotoxic effects on various cancer cell lines by inducing GPX4-mediated ferroptosis. Whether the Artesunate used in this study also has multiple derivatives that inhibit breast cancer progression remains unknown [11]. Both in vitro and clinical studies indicate that curcumin can influence multiple cellular signaling pathways, target molecular mechanisms in cancer cells, and induce apoptosis via intrinsic and extrinsic pathways. Its anti-cancer effects, which include anti-proliferative, pro-apoptotic, anti-metastatic, and anti-angiogenic actions, are linked to the inhibition of Nrf2 and the clinical applications of polyphenols as Nrf2 inhibitors [32]. Growing scientific evidence supports the therapeutic potential of many naturally derived molecules in cancer treatment.

### Limitations of natural products as cancer therapeutic candidates

Despite the therapeutic promise of natural products for cancer treatment, their clinical application faces several significant challenges. These include the difficulties in isolating and purifying active compounds in quantities sufficient for drug development, a lack of comprehensive understanding of their anti-cancer mechanisms, and pharmacokinetic and bioavailability issues that hinder pharmaceutical progress [33]. Moreover, natural products must undergo extensive testing and approval processes, which can be time-consuming and costly. There is also the risk of complex interactions with conventional cancer treatments, potentially leading to adverse effects or reduced efficacy. Additionally, environmental factors can cause variability in the chemical composition of natural products, leading to inconsistencies in their therapeutic effects [34,35]. There is still a long way to go for artesunate to be used as a clinical treatment drug for breast cancer. Firstly, there are still relatively few basic studies on artesunate in breast cancer. Artesunate can activate DNA damage and then induce cell-cycle arrest in breast cancer cells [36]. The killing effect of artesunate on breast cancer cell lines is mediated by specific EPH receptors and ephrin ligands [37]. Interestingly, the iron homeostasis maintained by transferrin and its receptor has been thoroughly studied in breast cancer treated with artesunate [38]. Whether the data from the above basic studies are sufficient to support further clinical research remains debatable. Secondly, the development of new formulations is still hindered. The toxicity identification and targeting ability of artesunate nanocapsules still need to be improved [39]. The biological properties of the new camptothecin-artesunate conjugate still need to be identified in terms of pharmacokinetics, pharmacology, and toxicology [40].

### The importance of mechanistic exploration of artesunate

As highlighted in the introduction, previous studies have examined Artesunate's effect on breast cancer cells. Being an FDA-approved drug for malaria, Artesunate has a promising potential to be repurposed for other medical uses, including breast cancer treatment. Therefore, more in-depth studies are urgently required to elucidate the detailed mechanisms underlying its inhibitory effects, which would provide stronger scientific support for its therapeutic potential and safety. A comprehensive literature review and RNAinter analysis were conducted to hypothesize the mechanistic pathway examined in this study.

## Summary of key findings from the current study

Based on a review of previous experiments, it was hypothesized that the inhibitory effects of Artesunate on breast cancer cells occur through its regulation of lncRNA TUG1. Our findings revealed that overexpression of lncRNA TUG1 promoted the proliferation of MCF-7 tumor cells, whereas Artesunate administration inhibited this proliferative effect. Knockdown of lncRNA TUG1 by siRNA reduced cell proliferation and this suppressive effect was further enhanced with Artesunate. Knockdown of lncRNA TUG1 also promoted cell apoptosis (as evidenced by TUNEL assay), and this pro-apoptotic effect was further amplified by Artesunate. Additionally, the knockdown of lncRNA TUG1 inhibited cell invasion (Transwell assay), and Artesunate further augmented this inhibitory effect. The same pattern was observed for cell migration (Wound Healing assay), where Artesunate enhanced the inhibitory effect of lncRNA TUG1 knockdown. Lastly, miR-145-5p was found to bind to lncRNA TUG1, inhibiting its transcriptional activation, and also to bind to HOXA5–3'-UTR, suppressing the transcriptional activation of HOXA5. These findings elucidate the function of the lncRNA/miR-145-5p/HOXA5 axis.

## Limitations of this study

Although this study has revealed the critical role of lncRNA TUG1 in the anti-tumor effects of Artesunate, the specific molecular mechanisms remain incompletely elucidated. For instance, whether lncRNA TUG1 exerts its functions through interactions with other miRNAs or proteins still requires further investigation. This study focused on the WNT/β-catenin signaling pathway, but whether it intersects with other signaling pathways, and whether it functions through positive or negative feedback regulation, remains unclear. Additionally, whether this study comprehensively represents the signaling pathway network is still a question. The study employed Artesunate at concentrations of 30μM, 120μM, and 480μM for experiments, but whether these doses are clinically feasible in vivo remains uncertain. High doses of Artesunate may induce toxicity in normal cells, potentially limiting its clinical application. Therefore, future research should further optimize the dosage of Artesunate and evaluate its safety and efficacy in vivo. Drug resistance is a common issue in cancer treatment, and future studies should assess whether Artesunate induces resistance and explore strategies to overcome it. The clinical application of Artesunate still faces numerous challenges, such as drug bioavailability, metabolic stability, and administration methods. Future research should explore the clinical translation potential of Artesunate and optimize its drug delivery systems to enhance its anti-tumor efficacy in vivo. Moreover, future studies should consider personalized treatment strategies, such as using genetic testing to screen patient populations sensitive to Artesunate, to improve treatment outcomes.

Artesunate is rapidly metabolized in vivo to dihydroartemisinin (DHA). DHA is further metabolized via glucuronidation by UDP-glucuronosyltransferases (primarily UGT1A9 and UGT2B7), forming α-DHA-β-glucuronide, which is the major metabolite of DHA. The elimination half-life of DHA is relatively short, ranging from 0.5 to 1.5 hours. In patients undergoing long-term (over 3 weeks) oral artesunate therapy, a 24.9% increase in the apparent oral clearance of DHA has been observed. Artesunate, administered orally, intravenously, or intramuscularly, is rapidly distributed throughout the body, with particularly high concentrations in the liver, spleen, and tumor tissues. However, artesunate may cause severe hepatotoxicity and bone marrow suppression when used in combination with dichloroacetate (DCA) or traditional Chinese herbs. At low doses, artesunate may exhibit anti-angiogenic effects, while at higher doses, it may directly kill tumor cells. Nonetheless, precise dose control is essential to avoid toxic side effects. When combined with other anticancer drugs (such as temozolomide or doxorubicin) or traditional Chinese herbs, artesunate may induce unpredictable toxic reactions. Although artesunate has demonstrated significant anticancer activity in vitro and in animal models, tumor cells may develop resistance through various mechanisms, such as alterations in DNA repair pathways, in clinical applications. Currently, the use of artesunate in cancer therapy is primarily based on small-scale clinical trials and case reports, and there is a lack of large-scale, multicenter randomized controlled trial data to validate its safety and efficacy.

The limitations of this study indeed lie in its focus on the role of Artesunate in regulating the malignant phenotypes of breast cancer cell lines through the LncRNA TUG1 target. Further exploration and research are needed to understand how Artesunate functions in vivo and how it can be applied in clinical settings. The study employed various concentrations of Artesunate in experiments, but it did not clearly specify whether these doses are clinically feasible in vivo. This indicates that the research was primarily conducted in vitro, lacking support from in vivo experimental data. While in vitro experiments are useful for preliminary exploration of drug mechanisms, their results may not fully reflect the true effects of the drug in the complex biological environment of the body, which involves various physiological and metabolic factors. High doses of Artesunate may induce toxicity in normal cells, potentially limiting its clinical application. However, the study did not provide data on the long-term toxicity of Artesunate. Long-term toxicity data are crucial for assessing the safety and potential side effects of a drug, especially in cancer treatment where patients may require prolonged medication. The study also did not explicitly mention the use of patient-derived tumor models (such as patient-derived xenograft models, PDX models). These models are better able to simulate the heterogeneity and complexity of human tumors and are important tools for evaluating drug efficacy and toxicity. If the study relied solely on standardized cell lines, its results may not adequately reflect the drug's effects in real patients. Therefore, the absence of patient-derived models could be a significant limitation of this study.

While this study has confirmed the breast cancer cell suppressive mechanism of artesunate through experimental validation, it did not exclude other potential mechanisms responsible for such inhibitory effects. A significant limitation of this research was the absence of single-cell sequencing analysis on MCF-7 cells post-treatment with artesunate. Conducting single-cell sequencing on both treated and control MCF-7 cells would have provided a comprehensive insight into the epigenetic alterations induced by artesunate, the target molecule. Although an underlying mechanism was proposed and validated with positive outcomes through literature review and RNAinter analysis, further in vitro assays and computational modeling are necessary to delve into the intricate 3D spatial interactions between artesunate and lncRNA TUG1. Moreover, future research is imperative to thoroughly elucidate the mechanistic role of artesunate in breast cancer treatment, thereby contributing more robust scientific evidence to support its potential for clinical trials shortly.

In conclusion, this study demonstrates that artesunate has the potential to modulate the malignant progression of breast cancer cells through the lncRNA TUG1/miR-145-5p/HOXA5 axis, offering promising evidence for its anti-cancer effects. However, the findings are primarily based on in vitro experiments, which may not fully capture the complexity of biological systems in vivo. To advance this research toward clinical application, further studies are imperative. These should include in vivo experiments to validate the efficacy and safety of artesunate in animal models, as well as clinical trials to assess its feasibility, optimal dosing, and long-term toxicity in human patients. Additionally, the use of patient-derived models, such as patient-derived xenografts (PDX), could provide more accurate insights into the drug's therapeutic potential in real-world scenarios. Addressing these gaps will be critical to translating these promising findings into effective and safe breast cancer therapies.

## Conclusion

In conclusion, artesunate has the potential to modulate the malignant progression of breast cancer cells through the lncRNA TUG1/miR-145-5p/HOXA5 axis, offering promising evidence for its future application in clinical trials for breast cancer treatment.

## Author contributions

**Conceptualization:** Chao Yang, Yunjiang Liu.

**Data curation:** Ziteng Zhang.

**Formal analysis:** Ziteng Zhang.

**Funding acquisition:** Chao Yang, Yunjiang Liu.

**Investigation:** Lingyun Gai.

**Methodology:** Lingyun Gai.

**Project administration:** Geng zhang.

**Software:** Yanshou Zhang.

**Supervision:** Chao Yang.

**Validation:** Kaiye Du, Chao Gao.

**Visualization:** Kaiye Du, Chao Gao.

**Writing – original draft:** Lingyun Gai.

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
