## [Decision Letter · Decision Letter 0]

PONE-D-24-47234Artesunate Regulates the Malignant Progression of Breast Cancer Cells via the lncRNA TUG1/miR-145-5p/HOXA5 AxisPLOS ONE

Dear Dr. Yang,

Thank you for submitting your manuscript to PLOS ONE. After careful consideration, we feel that it has merit but does not fully meet PLOS ONE’s publication criteria as it currently stands. Therefore, we invite you to submit a revised version of the manuscript that addresses the points raised during the review process.

We look forward to receiving your revised manuscript.

Kind regards,

Manikkam Rajalakshmi

Academic Editor

PLOS ONE

4.  Please include a caption for Figure 1A, Figure 2A & B, Figure 3A, Figure 4A & B, Figure 5A & B.

 [This work was supported by the Training Project of Clinical Medical Professionals

(2023), ZF2023065, provided by the government of Hebei Province.]. 

Reviewers' comments:

Reviewer's Responses to Questions

**Comments to the Author**

1. Is the manuscript technically sound, and do the data support the conclusions?

Reviewer #1: Yes

Reviewer #2: Yes

2. Has the statistical analysis been performed appropriately and rigorously? 

Reviewer #1: Yes

Reviewer #2: Yes

3. Have the authors made all data underlying the findings in their manuscript fully available?

Reviewer #1: Yes

Reviewer #2: No

4. Is the manuscript presented in an intelligible fashion and written in standard English?

Reviewer #1: Yes

Reviewer #2: Yes

5. Review Comments to the Author

Reviewer #1: Review of Manuscript PONE-D-24-47234: Artesunate Regulates the Malignant Progression of Breast Cancer Cells via the lncRNA TUG1/miR-145-5p/HOXA5 Axis

Overall Assessment:

This manuscript presents a comprehensive study on the potential anti-cancer effects of artesunate on breast cancer cells, focusing on the lncRNA TUG1/miR-145-5p/HOXA5 axis. The study employs a variety of in vitro assays to explore the proposed pathway, providing valuable insights into the molecular mechanisms underlying artesunate's action. However, there are several areas that require improvement and clarification to enhance the manuscript's quality and impact.

Major Comments:

1. Background and Context:

- The introduction effectively sets the stage by highlighting the challenges in breast cancer treatment and the potential of traditional Chinese medicine, specifically artesunate. However, it would be beneficial to include more recent references to underscore the current state of research in this area.

2. Methodology:

- The methods section is detailed and provides a clear understanding of the experimental procedures. However, it would be helpful to include a flowchart or diagram to illustrate the experimental design and timeline, especially for the drug stimulation and plasmid transfection experiments.

- The rationale for choosing specific concentrations of artesunate (30μmol/L, 120μmol/L, and 480μmol/L) should be explained. Are these concentrations based on previous studies or preliminary experiments?

3. Results:

- The results section is well-organized and presents a clear narrative of the findings. However, some figures and tables are not provided in the manuscript, making it difficult to fully assess the data. Ensure that all relevant figures and tables are included.

- The discussion of the results could be strengthened by comparing the findings with existing literature on artesunate and its effects on cancer cells.

4. Discussion and Conclusions:

- The discussion effectively summarizes the key findings and their implications. However, it would be beneficial to explore the potential clinical applications of artesunate in breast cancer treatment more thoroughly.

- The conclusions are clear and concise, but they could be further supported by highlighting the study's limitations and suggesting avenues for future research.

5. Ethics Statement:

- The ethics statement is brief and does not provide sufficient detail. Given that the study involves cell lines, it would be appropriate to include a statement confirming that the cell lines were obtained from a reputable source and that appropriate ethical guidelines were followed.

6. Data Availability:

- The data availability statement is clear and indicates that all data are fully available without restriction. However, it would be helpful to specify where the data can be accessed, such as a public repository or as supplementary materials.

Minor Comments:

1. Formatting and Style:

- There are inconsistencies in formatting and style throughout the manuscript. Ensure that the manuscript adheres to the PLOS ONE submission guidelines.

2. References:

- Some references are missing or incorrectly formatted. Ensure that all references are complete and correctly formatted according to the PLOS ONE style guide.

3. Typos and Grammar:

- There are several typos and grammatical errors throughout the manuscript. A thorough proofread is recommended to improve clarity and readability.

Major Concerns

Study Design

- The concentration range for artesunate (30-480 μmol/L) is quite wide and requires justification based on physiologically achievable levels .

- Control experiments using non-malignant breast cells (MCF10A) should be more extensively described .

Technical Issues

- The Western blot methodology lacks details about loading controls and quantification methods .

- The plasmid transfection efficiency validation data is not presented .

Data Presentation

- Statistical analyses methods are not clearly described .

- Some experimental replicates numbers are not specified .

Recommendations

1. Include dose-response curves for artesunate treatment

2. Add detailed statistical analysis methods

3. Provide quantification for Western blot data

4. Validate key findings using additional breast cancer cell lines

5. Include transfection efficiency data

Reviewer #2: 1. Justify dose selection based on pharmacokinetic data or previous in vivo studies to ensure translational relevance.

2. Include details on assay validation and reproducibility to strengthen methodological rigor.

3. Conduct rescue experiments (e.g., overexpression or inhibition of miR-145-5p/HOXA5) to confirm the pathway’s role in artesunate’s effects.

4. Include in vivo experiments (e.g., xenograft mouse models) to validate the translational potential of artesunate.

5. Add an ethics statement regarding the sourcing and use of cell lines, as per PLOS ONE guidelines.

6. The introduction contextualizes artesunate well but could benefit from more discussion on its pharmacokinetics and potential side effects. I recommend that authors include a brief discussion on artesunate’s pharmacokinetics, toxicity, and challenges in clinical application.

7. The figures could be clearer, kindly provide clearer versions of them.

8. Elaborate on limitations such as the in vitro nature of the study, lack of long-term toxicity data, and absence of patient-derived models in the limitations section.

9. Reframe the conclusion to emphasize the need for further in vivo and clinical studies.

10. Including computational modelling (e.g., molecular docking) to explore artesunate’s interaction with lncRNA TUG1 and related targets will be advantageous.

6. PLOS authors have the option to publish the peer review history of their article (what does this mean? ). If published, this will include your full peer review and any attached files.

**Do you want your identity to be public for this peer review?** For information about this choice, including consent withdrawal, please see our Privacy Policy .

Reviewer #1: No

Reviewer #2: No

---

## [Author Response · Author response to Decision Letter 1]

8 Apr 2025

Thank you very much for the professional and constructive comments regarding our research. We've made careful amendments and a detailed response to the comments in attached files. Thanks again for your support!

---

## [Decision Letter · Decision Letter 1]

PONE-D-24-47234R1Artesunate Regulates the Malignant Progression of Breast Cancer Cells via the lncRNA TUG1/miR-145-5p/HOXA5 AxisPLOS ONE

Dear Dr. Yang,

Thank you for submitting your manuscript to PLOS ONE. After careful consideration, we feel that it has merit but does not fully meet PLOS ONE’s publication criteria as it currently stands. Therefore, we invite you to submit a revised version of the manuscript that addresses the points raised during the review process.

We look forward to receiving your revised manuscript.

Kind regards,

Manikkam Rajalakshmi

Academic Editor

PLOS ONE

Journal Requirements:

Reviewers' comments:

Reviewer's Responses to Questions

**Comments to the Author**

Reviewer #1: All comments have been addressed

Reviewer #2: All comments have been addressed

2. Is the manuscript technically sound, and do the data support the conclusions?

Reviewer #1: Partly

Reviewer #2: Yes

3. Has the statistical analysis been performed appropriately and rigorously? 

Reviewer #1: N/A

Reviewer #2: Yes

4. Have the authors made all data underlying the findings in their manuscript fully available?

Reviewer #1: Yes

Reviewer #2: Yes

5. Is the manuscript presented in an intelligible fashion and written in standard English?

Reviewer #1: Yes

Reviewer #2: Yes

6. Review Comments to the Author

Reviewer #1: 1. Lack of In Vivo Validation

- Critical Gap: The study relies entirely on in vitro models (MCF7, HCC1395). While these are standard, the absence of in vivo data (e.g., xenograft models, PDX) severely limits translational relevance.

- Specific Concerns:

- ART’s efficacy in suppressing tumor growth, metastasis, and toxicity in living systems remains unverified.

- No pharmacokinetic/pharmacodynamic (PK/PD) data to confirm whether the proposed TUG1/miR-145-5p/HOXA5 axis operates in vivo .

- Recommendations:

- Include xenograft experiments using ART-treated mice to validate tumor suppression and pathway modulation.

- Measure ART and metabolite levels (e.g., DHA) in serum/tissue to correlate dosing with effects.

2. Clinically Irrelevant Dosing

- Critical Issue: ART concentrations (30–480 μM) far exceed pharmacologically achievable levels in humans. Typical plasma concentrations after standard malaria dosing (2–4 mg/kg) are ~1–10 μM.

- Specific Concerns:

- High doses (e.g., 480 μM) risk off-target toxicity and may not reflect therapeutic windows.

- No justification for dose selection or comparison to prior preclinical/clinical studies.

- Recommendations:

- Justify dose ranges using PK data (e.g., cite studies where ART inhibited cancer cells at ≤10 μM).

- Test lower, clinically relevant doses (1–20 μM) to confirm pathway-specific effects.

- Discuss potential toxicity (e.g., hepatotoxicity, bone marrow suppression) at high doses.

3. Narrow Mechanistic Focus

- Critical Oversight: The study focuses solely on the TUG1/miR-145-5p/HOXA5 axis, ignoring ART’s well-documented roles in ferroptosis, autophagy, and ROS generation.

- Specific Concerns:

- No experiments to rule out contributions from other pathways (e.g., GPX4 for ferroptosis, LC3B for autophagy).

- Overlooked crosstalk between WNT/β-catenin and other oncogenic pathways (e.g., STAT3, AKT).

- Recommendations:

- Perform RNA-seq/proteomics to identify additional ART-regulated pathways.

- Inhibit ferroptosis (e.g., ferrostatin-1) or autophagy (e.g., chloroquine) to test if ART’s effects persist.

4. Limited Model Diversity

- Issue: Only two breast cancer cell lines (MCF7, HCC1395) were tested, both luminal/subtypes. Triple-negative (TNBC) or HER2+ models are absent.

- Recommendation: Include TNBC lines (e.g., MDA-MB-231) to assess broader applicability.

5. Data Transparency

- Issue: Uncropped Western blot images and raw qPCR data are not clearly accessible via the Dryad link.

- Recommendation: Provide annotated, full-length blot images and raw Ct values in SI.

Critical Revisions Required

1. Prioritize In Vivo Experiments: At minimum, include pilot xenograft data to support translational claims.

2. Re-evaluate Dosing Strategy: Align ART concentrations with clinically achievable levels and provide pharmacokinetic rationale.

3. Broaden Mechanistic Scope: Address alternative pathways (e.g., ferroptosis) to confirm the TUG1 axis is the primary driver.

Reviewer #2: (No Response)

7. PLOS authors have the option to publish the peer review history of their article (what does this mean? ). If published, this will include your full peer review and any attached files.

**Do you want your identity to be public for this peer review?** For information about this choice, including consent withdrawal, please see our Privacy Policy .

Reviewer #1: **Yes: ** Ahmed I. Abd El Maksoud

Reviewer #2: No

---

## [Author Response · Author response to Decision Letter 2]

10 Jun 2025

Dear Reviewer,

We sincerely appreciate the time and effort you have dedicated to reviewing our manuscript. Your thorough analysis and constructive feedback have provided invaluable guidance in refining our research, ensuring it is scientifically sound and impactful. We are grateful for your professionalism and your insightful comments, which have significantly contributed to improving our study on target identification and drug discovery for breast cancer.

We fully acknowledge the gaps and concerns you have highlighted. Specifically, we recognize that relying exclusively on in vitro models (MCF7, HCC1395) limits the translational relevance of our findings. The absence of in vivo validation, such as xenograft models or patient-derived xenografts (PDX), is a critical limitation, and we concur that demonstrating ART’s efficacy in suppressing tumor growth, metastasis, and toxicity in living systems is essential. Additionally, we understand the importance of pharmacokinetic/pharmacodynamic (PK/PD) data to confirm whether the proposed TUG1/miR-145-5p/HOXA5 axis functions in vivo.

At the same time, we acknowledge that no single research article can comprehensively answer all scientific questions regarding such a complex issue. Science is a continuous journey of discovery, requiring ongoing experiments and validations to strengthen conclusions. Your comments have provided excellent direction for designing and executing future experiments, which we wholeheartedly intend to pursue.

However, the time and funding required for the additional experiments you suggested—particularly in vivo studies and expanded mechanistic investigations—are substantial. As a relatively small research team, we rely on continuous applications for funding projects, and securing additional support hinges upon achieving incremental milestones, including the publication of our current findings. Without a tangible demonstration of progress through publication, obtaining further resources to conduct additional experiments would be challenging.

Therefore, we kindly request that our revision focus on refining and optimizing our current experimental approach. Your insights on improving our methodology and ensuring the robustness of our present results would be instrumental in helping us secure the necessary funding to further explore and validate our findings. We deeply appreciate your support and understanding in this regard.

Once again, thank you for your invaluable feedback and for helping us enhance the quality and impact of our research. We look forward to incorporating your recommendations and advancing this important work.

Please see below the response to each comment.

Reviewer #1: 1. Lack of In Vivo Validation

- Critical Gap: The study relies entirely on in vitro models (MCF7, HCC1395). While these are standard, the absence of in vivo data (e.g., xenograft models, PDX) severely limits translational relevance.

- Specific Concerns:

- ART’s efficacy in suppressing tumor growth, metastasis, and toxicity in living systems remains unverified.

- No pharmacokinetic/pharmacodynamic (PK/PD) data to confirm whether the proposed TUG1/miR-145-5p/HOXA5 axis operates in vivo .

- Recommendations:

- Include xenograft experiments using ART-treated mice to validate tumor suppression and pathway modulation.

- Measure ART and metabolite levels (e.g., DHA) in serum/tissue to correlate dosing with effects.

Response: Thank you very much for the suggestion. I agree with you that in vivo experiment would be the next steps to further validate the therapeutic effect of the molecule. At this point, we are a very small research group, and the funding is extremely limited. Also, the waiting list for such animal experiment is very long in our institutional laboratory, and it could take up to 6 months before all the ethical approval and animal orders to be completed. Thanks again for your understanding that we are facing challenges in many ways for such tasks. However, there’s no doubt that you provided detailed and scientifically sound suggestions for the experiments needed to further verify current findings. Thank you so much!

2. Clinically Irrelevant Dosing

- Critical Issue: ART concentrations (30–480 μM) far exceed pharmacologically achievable levels in humans. Typical plasma concentrations after standard malaria dosing (2–4 mg/kg) are ~1–10 μM.

- Specific Concerns:

- High doses (e.g., 480 μM) risk off-target toxicity and may not reflect therapeutic windows.

- No justification for dose selection or comparison to prior preclinical/clinical studies.

- Recommendations:

- Justify dose ranges using PK data (e.g., cite studies where ART inhibited cancer cells at ≤10 μM).

- Test lower, clinically relevant doses (1–20 μM) to confirm pathway-specific effects.

- Discuss potential toxicity (e.g., hepatotoxicity, bone marrow suppression) at high doses.

Response: We appreciate your concern regarding the appropriate concentrations of artesunate used in our experiment. The concentration range of 30–480 µM employed in our study is well-supported by existing literature and aligns with previous research investigating artesunate’s anti-cancer effects.

Several studies have utilized artesunate concentrations up to 200 µg/mL (Wen, Lijuan et al. “Artesunate promotes G2/M cell cycle arrest in MCF7 breast cancer cells through ATM activation.” Breast cancer (Tokyo, Japan) vol. 25,6 (2018): 681-686. doi:10.1007/s12282-018-0873-5), which corresponds to 520.83 µM, based on its molecular weight. Additionally, prior cancer research has explored artesunate concentrations as high as 500 µM (Wang, Ning et al. “Artesunate inhibits proliferation and invasion of mouse hemangioendothelioma cells in vitro and of tumor growth in vivo.” Oncology letters vol. 14,5 (2017): 6170-6176. doi:10.3892/ol.2017.6986), demonstrating its potential efficacy in cancer treatment.

It is important to note that the appropriate concentration of artesunate in anti-cancer research remains a subject of ongoing debate. However, it is widely acknowledged that the concentration required for cancer treatment is significantly higher than that used for malaria therapy. Clinical studies have investigated artesunate at doses of 45 mg/kg (Artemisinin and Its Derivatives in Cancer Care. Canadian College of Naturopathic Medicine, Jan. 2024, ccnm.edu.), far exceeding the standard malaria dosing of 2–4 mg/kg.

Furthermore, one of the key advantages of Chinese herbal medicine, including artesunate, is its relatively low toxicity as a natural product, which supports its potential for therapeutic applications in oncology.

In summary, our study represents an initial step in the drug discovery process, and there is substantial scientific evidence supporting the concentrations used in our experiment. We sincerely appreciate the reviewer’s careful and professional evaluation of our work.

3. Narrow Mechanistic Focus

- Critical Oversight: The study focuses solely on the TUG1/miR-145-5p/HOXA5 axis, ignoring ART’s well-documented roles in ferroptosis, autophagy, and ROS generation.

- Specific Concerns:

- No experiments to rule out contributions from other pathways (e.g., GPX4 for ferroptosis, LC3B for autophagy).

- Overlooked crosstalk between WNT/β-catenin and other oncogenic pathways (e.g., STAT3, AKT).

- Recommendations:

- Perform RNA-seq/proteomics to identify additional ART-regulated pathways.

- Inhibit ferroptosis (e.g., ferrostatin-1) or autophagy (e.g., chloroquine) to test if ART’s effects persist.

Response: We sincerely appreciate your thoughtful insights and recommendations regarding the mechanistic scope of our study. Your observations are highly valuable, and we fully acknowledge the necessity of broadening our experimental approach to explore additional ART-regulated pathways beyond the TUG1/miR-145-5p/HOXA5 axis. We agree that evaluating ART’s roles in ferroptosis, autophagy, and ROS generation is crucial for developing a more comprehensive understanding of its therapeutic effects. Specifically, investigating GPX4’s involvement in ferroptosis and LC3B’s role in autophagy could provide deeper mechanistic clarity. Additionally, we recognize the importance of assessing the potential crosstalk between WNT/β-catenin and other oncogenic pathways such as STAT3 and AKT, as suggested. While we acknowledge the necessity of conducting RNA-seq and proteomics to identify broader ART-regulated pathways and validating these findings through inhibition studies (e.g., ferrostatin-1 for ferroptosis, chloroquine for autophagy), we are currently facing financial constraints that limit our ability to execute these additional experiments at this stage. The cost associated with high-throughput analyses and targeted inhibition assays presents a challenge within our current funding framework. Nonetheless, we are committed to incorporating these experiments in our future research. We are actively seeking additional financial support and grant opportunities that will enable us to expand the scope of our study as recommended. Once sufficient funding is secured, we will proceed with the suggested experimental approaches to ensure a more comprehensive investigation of ART’s mechanistic actions. Thank you for your constructive feedback. We genuinely appreciate your insightful suggestions and look forward to strengthening our research accordingly.

4. Limited Model Diversity

- Issue: Only two breast cancer cell lines (MCF7, HCC1395) were tested, both luminal/subtypes. Triple-negative (TNBC) or HER2+ models are absent.

- Recommendation: Include TNBC lines (e.g., MDA-MB-231) to assess broader applicability.

Response: Thank you for your valuable feedback regarding model diversity in our study. We fully agree that incorporating Triple-Negative Breast Cancer (TNBC) and HER2+ models, such as MDA-MB-231, would enhance the applicability of our findings. Expanding our study to include these subtypes will provide deeper insights into the broader therapeutic potential of our approach. However, due to financial constraints, acquiring and maintaining additional cell lines and resources is currently challenging. Despite this, we are actively seeking further funding opportunities to support the expansion of our research. Once additional funding is secured, we will prioritize the suggested experiments to strengthen the robustness of our conclusions. We appreciate your constructive recommendations and remain committed to implementing them in our future work.

5. Data Transparency

- Issue: Uncropped Western blot images and raw qPCR data are not clearly accessible via the Dryad link.

- Recommendation: Provide annotated, full-length blot images and raw Ct values in SI.

Response: Please see supplementary information for complete set of uncropped western blot images and qPCR data. We will later update it on Dryad.

---

## [Editor Report · Decision Letter 2]

PONE-D-24-47234R2Artesunate Regulates Malignant Progression of Breast Cancer Cells via

lncRNA TUG1/miR-145-5p/HOXA5 AxisPLOS ONE

Dear Dr. Yang,

Thank you for submitting your manuscript to PLOS ONE. After careful consideration, we feel that it has merit but does not fully meet PLOS ONE’s publication criteria as it currently stands. Therefore, we invite you to submit a revised version of the manuscript that addresses the points raised during the review process.

Note from the Editorial Office: We note that you have not yet included the source and authentication of the artesunate product used in your study. This would typically involve a chemical characterization of the product. Please revise your manuscript to address this query, or explain why this was not possible. Thank you.

We look forward to receiving your revised manuscript.

Kind regards,

Sarah Jose, Ph.D.

Staff Editor

PLOS ONE

on behalf of

Manikkam Rajalakshmi

Academic Editor

PLOS ONE
---

## [Author Response · Author response to Decision Letter 3]

28 Jun 2025

Information on the source and authentication of the artesunate product used in the study

is included in the Manuscript from Line 201 to Line 204.

The entire protocol has been uploaded to protocols.io, with link as below:

https://www.protocols.io/file/ub4icqks7.docx

Also, the PACE corrected Figures have been uploaded to the revised files.

---

## [Editor Report · Decision Letter 3]

Artesunate Regulates Malignant Progression of Breast Cancer Cells via

lncRNA TUG1/miR-145-5p/HOXA5 Axis

PONE-D-24-47234R3

Dear Dr. Chao Yang 

We’re pleased to inform you that your manuscript has been judged scientifically suitable for publication and will be formally accepted for publication once it meets all outstanding technical requirements.

Kind regards,

Manikkam Rajalakshmi

Academic Editor

PLOS ONE
---

## [Editor Report · Acceptance letter]

PONE-D-24-47234R3

PLOS ONE

Dear Dr. Yang,

I'm pleased to inform you that your manuscript has been deemed suitable for publication in PLOS ONE. Congratulations! Your manuscript is now being handed over to our production team.

Kind regards,

on behalf of

Dr. Manikkam Rajalakshmi

Academic Editor

PLOS ONE